# Comparing Control Intervention Scenarios for Raccoon Rabies in Southern Ontario between 2015 and 2025

**DOI:** 10.3390/v15020528

**Published:** 2023-02-14

**Authors:** Emily Sohanna Acheson, François Viard, Tore Buchanan, Larissa Nituch, Patrick A. Leighton

**Affiliations:** 1Research Group on Epidemiology of Zoonoses and Public Health (GREZOSP), Faculty of Veterinary Medicine, University of Montréal, 3200 Rue Sicotte, Saint-Hyacinthe, QC J2S 2M2, Canada; 2Centre de Recherche en Santé Publique, l’Université de Montréal et du CIUSSS du Centre-Sud-de-l’île-de-Montréal (CReSP), Montréal, QC H3N 1X9, Canada; 3Public Health Risk Sciences Division, National Microbiology Laboratory, Public Health Agency of Canada, 3200 rue Sicotte, Saint-Hyacinthe, QC J2S 2M2, Canada; 4Wildlife Research and Monitoring Section, Ontario Ministry of Natural Resources and Forestry, Trent University, 2140 East Bank Drive, Peterborough, ON K9L 1Z8, Canada

**Keywords:** agent-based model, control interventions, geographic information systems, Python, *Rabies lyssavirus*, Ontario, outbreak, raccoon

## Abstract

The largest outbreak of raccoon rabies in Canada was first reported in Hamilton, Ontario, in 2015 following a probable translocation event from the United States. We used a spatially-explicit agent-based model to evaluate the effectiveness of provincial control programs in an urban-centric outbreak if control interventions were used until 2025, 2020, or never used. Calibration tests suggested that a seroprevalence of protective rabies antibodies 2.1 times higher than that inferred from seroprevalence in program assessments was required in simulations to replicate observed raccoon rabies cases. Our simulation results showed that if control interventions with an adjusted seroprevalence were used until 2025 or 2020, the probability of rabies elimination due to control intervention use was 49.2% and 42.1%, respectively. However, if controls were never used, the probability that initial rabies cases failed to establish a sustained outbreak was only 18.2%. In simulations where rabies was not successfully eliminated, using control interventions until 2025 resulted in 67% fewer new infections compared to only applying controls until 2020 and in 90% fewer new infections compared to no control intervention use. However, the model likely underestimated rabies elimination rates since we did not adjust for adaptive control strategies in response to changes in rabies distributions and case numbers, as well as extending control interventions past 2025. Our agent-based model offers a cost-effective strategy to evaluate approaches to rabies control applications.

## 1. Introduction

Rabies is a vaccine-preventable zoonotic disease transmissible through the saliva of an infected mammal and is nearly 100% fatal once symptoms appear [1]. In Ontario, Canada, the Ontario Ministry of Natural Resources and Forestry (MNRF) has led rabies control programs in the province for over 33 years. Prior to 1979, the province had earned the distinction of being the rabies capital of North America due to its high numbers of human post-exposure treatments (PETs) and domestic animal cases [2]. The raccoon rabies virus variant is widespread throughout the eastern seaboard of the United States [3]. The raccoon (*Procyon lotor*) is a generalist species that has adapted to both urban and rural environments in Southern Ontario [4]. Its ability to thrive in high-density urban settings, denning and feeding in and around human dwellings and other buildings, poses a public and animal health risk. This risk was heightened in December 2015 when the largest outbreak of raccoon rabies in Canada was first reported in Hamilton, Ontario, following a probable cross-border translocation from the United States [5]. In response, provincial control programs targeted Hamilton and the surrounding regions to limit potential public and animal health impacts and attempt to eliminate the disease [6].

Wildlife vaccination can be a crucial component of zoonotic disease control. Wildlife vaccination against raccoon rabies in Ontario includes a variety of methods, such as trap-vaccinate-release (TVR) of wildlife with an intramuscular vaccine and oral vaccine bait distribution from aircraft, vehicles, and bait stations to optimize animal targeting in urban and rural settings. However, disease control programs must account for the spatial heterogeneity of natural and built-up environments, the behavioural patterns of the target animals, the epidemiological traits of the pathogen, and the effectiveness of the control interventions. Field studies are often impractical and labour-intensive to test the effectiveness of vaccination strategies or lack of intervention on zoonotic diseases under these varying conditions. Agent-based modelling provides another avenue for evaluating intervention strategies, incorporating details of host and pathogen biology and dynamics, control interventions, and the spatial complexities of the study area. By simulating individual agents (in this case, raccoons) over a landscape across a specified time period, agent-based models (ABMs) can account for variability in host populations, pathogen distributions, and control applications, across space and through time. For example, Rees et al. [7] evaluated the effectiveness of vaccine barrier strategies for raccoon rabies control, varying barrier widths and seroprevalence over spatially heterogeneous habitats. Advancements in computational resources have created an opportunity to increase the spatial resolution of host habitats, the area of study, and the number of modelled agents [8].

In the current study, we used a spatially-explicit ABM to simulate spatio-temporal raccoon rabies dynamics across Southern Ontario in the context of the 2015 rabies outbreak in Hamilton. We aimed to model rabies control interventions used from 2015 to 2020, as well as those proposed for use until 2025, and compared the severity of rabies outbreaks in three scenarios: (one) using control interventions until 2025, (two) using control interventions only until 2020, and (three) not using control interventions at all.

## 2. Materials and Methods

### 2.1. Agent-Based Model

We used the open-source Python library called SamPy (Stochastic Agent-based Modelling with Python) ([9], preprint) to model the raccoon population in Southern Ontario. Using SamPy to model raccoon rabies in Southern Ontario is comparable to using the Ontario Rabies Model (ORM; [10]; see Appendix A for a comparison of ORM and SamPy modelling of raccoon rabies spread rates), another spatially explicit agent-based model that has been used and validated extensively to model rabies in raccoons in Southern Ontario [7,11,12]). However, SamPy provides faster processing speeds and flexibility for further development ([9], preprint). We used the Overview, Design concepts, and Details (ODD) standard protocol for describing ABMs [13] to describe our use of SamPy.

### 2.2. Overview

#### 2.2.1. Purpose

In this study, SamPy was used as a tool to simulate the dynamics of raccoon populations and raccoon rabies across Southern Ontario in order to estimate the efficacy of rabies control interventions applied across the study region from 2015 to 2020. The model was also used to predict the efficacy of these interventions if they continued to be applied until 2025. Simulations were run only until 2025 as this was the year rabies managers had predicted for raccoon rabies elimination in southern Ontario (MNRF, Peterborough, Ontario, Canada, 2022). We calculated the number of rabies infections in raccoons per week from 2015 to 2025 across an initially disease-free area and compared scenarios where control interventions were or were not applied after 2015. Finally, we compared these results with a scenario where no control interventions were applied.

#### 2.2.2. Entities, State Variables, and Scales

SamPy is spatially explicit and models agents across a fixed user-defined landscape. For the purposes of this study, the agents were individual raccoons and raccoon families (a mother raccoon and her dependent offspring), and the landscape consisted of spatially contiguous hexagonal grid cells that composed the study area. Each individual had a unique identity number, along with sex, parents’ identity numbers, and geographical position. Each raccoon agent was either a dependent offspring (less than 20 weeks of age), a juvenile (at or over 20 weeks and under 75 weeks of age), or an adult (at or over 75 weeks of age). The study area was comprised of 28,564 grid cells, each with an area of 10 km^2^. Each grid cell had a carrying capacity, K, which represented the average number of agents at Week 30 (the end of July) sustained by the habitat in that grid cell (Figure 1) [7] (for details regarding calculations of K values, see Appendix A). For each model time step, the total number of raccoons and the number of newly infected adult raccoons per grid cell were calculated.

#### 2.2.3. Process Overview and Scheduling

Discrete time steps of one week were used for modelling processes. Some parameters were only applied at a specific week (e.g., female raccoons gave birth each year at Week 18), while other parameters would be applied across multiple weeks (e.g., juvenile male raccoons may disperse at any week between Weeks 38 and 43) (Table 1). The timing and frequency of processes were determined by the sex and age of agents. Model processes were categorized as (1) population dynamics, (2) disease, and (3) disease control. Three model scenarios were tested, with each model running weekly from 1 January 2015 to 31 December 2025 (i.e., Week 1 of 2015 to Week 52 of 2025, or 572 weeks in total). In the first model scenario, control interventions were applied from 2015 to 2025. In the second model scenario, control interventions were only applied through 2020. In the third model scenario, control interventions were not applied at all.

Population dynamics were stochastic and included reproduction, dispersal, and non-disease-related mortality. Mating pairs were created by randomly selecting one male and one female at or over 52 weeks of age in the same grid cell, and the number of resulting offspring was determined by a probability distribution. Juvenile and adult raccoon agents were allowed to disperse each year (i.e., move from the grid cell they currently occupy). The range of weeks within which a raccoon could disperse was dependent on its age and sex. Dispersal distance and direction were random. Each raccoon agent dispersed within one week and moved to its destination grid cell without interacting with other raccoons occupying any grid cells in between. Mother raccoons (i.e., female raccoon agents with dependent offspring under 20 weeks of age) did not disperse. Once dependent offspring reach the age of independence (i.e., 20 weeks of age), they may disperse and contract and transmit rabies. Each raccoon had a risk of dying each week, where mortality risk was dependent on age, sex, and the ratio of raccoons currently occupying a grid cell to the carrying capacity of that grid cell. Raccoon agents could also die from rabies each week. Raccoon agents could live for a maximum of eight years.

Raccoon agents exist in one of four states with respect to disease (Susceptible, Incubating, Infectious or Immune [7]), analogous to the Susceptible (S), Exposed (E), Infected (I) and Recovered (R) disease states used in classic compartmental epidemiological models. Susceptible individuals that become infected with rabies enter the Incubating state, and subsequently transition to the Infectious state according to a field-derived distribution of rabies incubation periods in raccoons (Table 1). Raccoon agents remain Infectious for one week, during which they can infect Susceptible individuals in their occupied cell or adjacent cells, before dying from rabies or entering the Immune state.

In our simulations, rabies infection was introduced by infecting five percent of randomly selected individuals [7] in each of four grid cells in Week 47 (the third week of November) of 2015. Week 47 was used after conducting calibration tests to estimate the start week of initial infections (see Appendix A for details). These grid cells spatially aligned with the georeferenced location of the four earliest raccoon rabies cases in Southern Ontario, which were detected in early December 2015. At each time step, raccoons over 20 weeks of age had a 22.2% chance of leaving their current grid cell [7,10] to enter a randomly selected neighbouring grid cell. Activity patterns of infected and non-infected individuals were the same, given that behavioural changes often associated with rabies infection generally only manifest themselves during the final acute stage of infection (represented here by the Infectious state), which has a duration of a single time step in our model [7]. A neighbouring grid cell was defined as a grid cell that shared a vertex with the current grid cell. In contrast to dispersion, raccoons that left their current grid cell to interact with raccoons in a neighbouring grid cell could do so at every time step as opposed to specific weeks and were returned to their original grid cell at the end of the time step. Every infectious individual could infect every susceptible individual occupying the same grid cell with a transmission probability of 3.5% (Appendix A), which has been used previously in agent-based modelling of raccoon rabies in Southern Ontario (Table 1; [7,10]). Due to the existence of documented naturally-occurring rabies antibodies in raccoons [14,15], the disease-induced mortality rate was set at 95% [7]. Here, we assumed that naturally occurring rabies antibodies in raccoons are protective and remain so throughout the lifetime of the animal.

Disease control methods included hand, fixed-wing, and helicopter baiting, as well as bait stations and TVR. The week(s), month, and year, as well as the georeferenced location of each control intervention applied in Southern Ontario from 2015 to 2020, were provided by the Ministry of Natural Resources and Forestry (MNRF). In this study, each disease control method was applied within one week per year per grid cell. In the case of multiple weeks of application, we used the median week. For example, if a given grid cell underwent hand baiting from August 1st to August 21st of 2015 (i.e., Weeks 31 to 33 of 2015) and then underwent helicopter baiting on 30 September 2015 (i.e., Week 39 of 2015), the grid cell would undergo hand baiting on Week 32 and helicopter baiting on Week 39 in our study. In the case of multiple interventions being applied in the same cell in the same week and year, the intervention with the highest associated seroprevalence was applied. Serological testing of field-captured animals conducted by the Canadian Food Inspection Agency and MNRF between 2016 and 2020, combined with calibration tests on seroprevalence (Appendix A), helped estimate the proportion of the raccoon population immunized by each control method. The seroprevalence represents the percentage of seropositive raccoons living in an area subjected to a disease control eight weeks following control intervention applications. Serology tests were conducted on raccoons in Southern Ontario six to eight weeks after the end of disease control applications, and results were based on a titre of 0.5 international units per ml [16]. Model calibration tests estimated that a level of protective immunity 2.1 times higher than the level of seropositivity observed in raccoons in the field resulted in epidemic curves most closely matching the observed rabies cases associated with the Hamilton outbreak from 2015–2020 (Appendix A). The only exception to the multiplication factor was TVR, which was maintained at its measured seroprevalence of 60% due to higher confidence in successful vaccination (e.g., in 2018, 92.9% of raccoons ear tagged from the TVR program were seropositive, ~65% of the raccoon population was estimated to be captured and vaccinated, therefore resulting in approximately 60% of the raccoon population conferred with a protective titer and successfully seroconverted; MNRF unpublished).

Proposed control interventions from 2021 to 2025 were also provided by the MNRF, and the same methods were used for calculating the median week of application and seroprevalence (i.e., the same seroprevalence used for each control intervention type were applied across all model scenarios). SamPy determined whether a raccoon was immune if (1) the raccoon agent survived rabies infection (5% of raccoons in our simulations survived) or (2) during a time step when a control intervention was being applied to a grid cell, SamPy randomly selected raccoon agents in the grid cell that corresponded to the target seroprevalence (i.e., percentage of raccoons receiving a vaccine). Infected raccoon agents could not receive a vaccine. Raccoon agents that survived rabies infections were immune for life. Raccoon agents receiving a vaccine were immune for 156 weeks (i.e., three years; see Appendix A for details on vaccine application and duration in SamPy).

### 2.3. Design Concepts

#### 2.3.1. Emergence

Raccoon population and disease dynamics, including the number of non-diseased and diseased raccoon agents per grid cell and their spatial distribution across the study area, emerge as the result of individual non-disease- and disease-related behaviours and traits that act on each raccoon. These behaviours and traits, such as sex- and age-specific dispersal, reproduction, non-disease- and disease-related mortality, interactions with raccoon agents in neighbouring grid cells, and incubation period, are the result of probabilities that are decided upon using a random number generator. For example, if an infected raccoon agent enters a grid cell with 10 susceptible raccoon agents and has a transmission probability of 3.5%, SamPy uses a random number generator to create a distribution and select a random number for each of the 10 interactions to determine whether or not transmission occurs. If the random number selected had a value at or below 3.5% of the numbers generated, transmission occurred between the infected and the susceptible raccoon agent. Infections also spread or waned as a result of disease control strategies, which resulted in the emergence of vaccinated raccoon agents.

#### 2.3.2. Interactions

Individual raccoon agents interacted either through mating or through disease transmission. During mating, one male and one female at or over 52 weeks of age occupying the same grid cell were randomly selected, and the number of resulting offspring was determined by a probability distribution. Dependent offspring then remained with their mother until the age of independence (>=20 weeks of age). During disease transmission, an infected raccoon agent had the opportunity to interact with every other raccoon agent occurring in the same grid cell at each time step. Only raccoon agents at or over the age of independence could transmit rabies. For each interaction, the infected raccoon agent had a probability of 3.5% of infecting every susceptible individual [7,17]. These interactions would occur if (one) one or more susceptible raccoon agents entered the grid cell of the infected raccoon agent during weekly interactions with neighbouring grid cells (where each raccoon agent over 20 weeks of age had a 22.2% chance of interacting with a neighbouring grid cell at each time step) or during dispersion, or (two) the infected raccoon agent entered a grid cell with one or more susceptible raccoon agents during weekly interactions with neighbouring grid cells or during dispersion. During dispersion, raccoon agents only interact with other raccoons in the destination grid cell and not those occupying grid cells in between.

#### 2.3.3. Stochasticity

Raccoon behaviours and traits such as non-disease annual mortality rates, litter size, dispersal distance, dispersal direction, and incubation period are stochastic processes (Table 1). In SamPy, stochastic processes are determined by a random number generation used from discrete probability distributions.

#### 2.3.4. Collectives

All raccoon agents within a grid cell are collectively affected by the same habitat and disease controls at each time step. Mother raccoon agents and their offspring also act as collectives while offspring are younger than 20 weeks of age (i.e., the age of independence) (Table 1). If the mother dies while her offspring are under 20 weeks of age, all of her offspring will die at the same time step.

#### 2.3.5. Observation

We recorded weekly data for each grid cell across the study area using an omniscient point of view. For the raccoon population building process, we calculated the total number of agents per grid cell per week (including dependent offspring). We also recorded the ID of each individual, the ID of the mother and father of that individual, its gender, whether it was pregnant, and which grid cell it inhabited at the end (i.e., the last week) of the simulation. For the disease and vaccination processes we calculated, per week and per grid cell, the number of agents, the number of deaths, the number of new rabies infections, and the number of vaccinated individuals.

### 2.4. Details

#### 2.4.1. Initialization

We created a model landscape of spatially contiguous 10-km^2^ hexagonal grid cells across the study area of Southern Ontario, resulting in a landscape of 28,564 grid cells. Carrying capacity (K) values were determined from field study literature of raccoons in Ontario [4], Québec [18], and the eastern United States [19], combined with expert opinion (T. Buchanan and L. Nituch). Values were usually provided as a raccoon density index (RDI) value, which specified the number of raccoon agents expected to be found in each habitat type per km^2^ (Appendix A). Since the hexagonal grid cells used in our study area were 10 km^2^, each RDI value was multiplied by 10 to calculate the predicted K value.

The model landscape was seeded with five mating pairs in five different grid cells across the landscape and run for 150 model years, where population growth stabilized. Due to the stochastic nature of SamPy, we conducted 1000 population-building trials. Each trial served as the starting population for the rabies introduction and vaccination simulations.

#### 2.4.2. Input

The input values for the raccoon population and disease dynamics parameters (Table 1) were obtained from field studies of North American raccoon and raccoon rabies systems, such as those detailing raccoon density indices [4,19], as well as previous simulations in the ORM [7]. The inputs were used in calibration tests along with varying start weeks and seroprevalence to ensure epidemic curves produced by SamPy aligned with that from real raccoon rabies infections in Southern Ontario collected by the MNRF between 2015 and 2020. These tests were also run to ensure SamPy responded to input parameter changes as expected. The model was run from January 1st of 2015 (i.e., Week 1 of 2015), rabies was introduced in Week 47 in the four cells that spatially aligned with the earliest four raccoon rabies cases detected in December 2015, and was run through the 52nd week of 2025. Control interventions (or lack of control intervention) were applied in the three model scenarios at 1000 runs per scenario, using the same 1000 population outputs from the population building process as inputs for each scenario.

## 3. Results

Raccoon rabies was most likely to be eliminated when control interventions were applied until 2025 and least likely to be eliminated when controls were not used at all (Figure 2 and Figure 3). When control interventions were applied until 2025 (Scenario 1), 67.4% of simulations predicted rabies elimination from the study region by 2025. When control interventions were halted after 2020 (Scenario 2), 60.3% of simulations predicted rabies elimination from the study region. For each of the three scenarios, 18.2% of the simulations predicted rabies would die out one week after its introduction when controls were not yet applied. Therefore, Scenarios 1 and 2 predicted rabies elimination in 49.2% and 42.1% of simulations, respectively, due to control intervention use (Figure 2 and Figure 3). When control interventions were not used at all (Scenario 3), the only simulations predicting rabies elimination were those when rabies died out one week after its introduction.

When evaluating median monthly rabies infections, we grouped simulations within each scenario as either (one) rabies was eliminated or (two) rabies was not eliminated by the end of 2025. For Scenarios 1 and 2, epicurves for simulations predicting rabies elimination were nearly identical and predicted no new rabies infections by February of 2019 and September of 2018, respectively, with five cases or less per month as of April 2018 (Figure 2A,C). The predicted number of new infections peaked for both scenarios in June of 2016 with 156 and 152 cases, respectively, with a total of 1441 and 1401 new infections across the 11-year period. For simulations predicting rabies would not be eliminated, epicurves all showed seasonal fluctuations in new infections, with peaks in the winter months (November through January). In both Scenarios 1 and 2, seasonal fluctuations in new infections became consistent (i.e., reaching similar yearly peak levels) starting in 2023. New infections peaked at 3804.5 in December of 2022 and 13,293 cases in January of 2025, respectively. Seasonal fluctuations in Scenario 3 became consistent starting in 2019, reaching a maximum of 15,725.5 cases in November 2023. Across the 11-year period, Scenarios 1, 2, and 3 totaled 86,881, 262,595, and 872,882.5 new infections, respectively.

Maps of the predicted geographic distribution of raccoon rabies for these scenarios were also made using the same simulation groupings (Figure 3). For each 10-km^2^ grid cell, we calculated the percentage of simulations predicting at least one raccoon rabies case in that cell between November 2015 and December 2025. The spatial extents of rabies infections were similar to the simulations in Scenarios 1 and 2, where the disease was eliminated. For both scenarios, ≥50% of simulations predicted rabies to spread within 23 km of the first four seeded cases in the Hamilton region, with the likelihood of spreading beyond this distance decreasing in all directions (Figure 3A,C). Rabies spread, at most, 74 km from Hamilton in any direction with the greatest spread towards the west.

For the simulations where rabies was not eliminated, Scenario 3 showed the greatest spatial spread and the highest likelihood of reaching nearly every cell in the study region below the boreal forest line. By 2025, the disease was predicted to spread as far as 250 km north and 350 km south of Hamilton, with ≥80% chance of reaching nearly every cell in this area (Figure 3E). The spatial spread and likelihood of spread decreased as the number of years of control applications increased. Scenario 1 showed the most limited spatial spread in simulations where rabies was not eliminated, with ≥50% chance of spreading within 36 km west, south, or east of Hamilton but 170 km north of Hamilton (Figure 3B). By comparison, Scenario 2 also predicted a ≥50% chance of spreading 170 km north of Hamilton, but predicted a more even spatial spread in all other directions, with ≥50% chance of spreading between 100 and 120 km west, south, or east (Figure 3D).

## 4. Discussion

We compared scenarios where rabies control interventions were applied for five or ten years after the 2015 raccoon rabies outbreak in Southern Ontario, as well as a scenario where no controls were applied. We found that the longer rabies control interventions were applied over time, the higher the likelihood that raccoon rabies would be eliminated from Southern Ontario. Even if rabies was not successfully eliminated by the end of 2025, using control interventions for 10 years resulted in 67% fewer total new infections compared to only using them for five years and resulted in 90% fewer total new infections compared to no control intervention use. Relative to no control intervention use, using control interventions for the first five years reduced the total number of new infections by nearly 70% by the end of 2025. We also found that there was nearly a one in five chance of rabies dying out within the first week of its introduction, but if it was successfully transmitted after that first week, it could not be eliminated without the use of control interventions.

The outbreak of raccoon rabies in Southern Ontario is the first outbreak in Canada to be detected in a densely populated urban area [6]. Compared to rural areas, (sub)urban areas support greater densities of raccoons. We replicated these higher densities in our synthetic landscape of Southern Ontario, which ultimately played a role in the spatial distribution of infections. In simulations where rabies was not eliminated by 2025, the disease was more likely to head north through the Greater Toronto Area (GTA) and taper off ~170 km north of Hamilton, where the habitat became less suitable for raccoons (Figure 3B,D). Field studies indicate there are ~18 raccoons per km^2^ in these urban areas [18], with higher densities (up to 21 raccoons per km^2^) in environments with combinations of deciduous forest and either agricultural fields and/or developed environments [19,20]. These field study findings were reflected in the K values assigned to each grid cell in the model landscape, along with low K values in areas with coniferous forests (~2.9 raccoons per km^2^) [18]. The low habitat suitability of coniferous forests may explain the maximum northern extent of rabies in all three scenarios (Figure 3). The potential spatial expansion of raccoon rabies through the GTA emphasizes the importance of control intervention placement just north of Hamilton, where a spatial bottleneck of high K values provides passage for the disease into high-density urban raccoon habitats. When control interventions are not used at all, the widest spatial spread of rabies can be seen both north and south of the first detected cases. While the predicted geographic distribution of rabies for Scenario 3 did not reach the very southern points of Ontario, the disease would almost certainly reach those areas if the modelling period were extended beyond 2025 since carrying capacity values increase towards the (sub)urban areas in and around Windsor.

Our model predicted a seasonal peak in new infections in the winter months (November to January). In detected cases in Southern Ontario, peaks mainly occurred in the summer months, except at the beginning of the outbreak. The month with the highest detected number of raccoon rabies infections was in December 2015 with 23 cases (Appendix A). The contradiction between the predictions of peaks in our model compared to detected cases in the field may be due to a decreased ability to detect cases in winter months. Our model replicates dispersal patterns in raccoons, with the dispersal and interactions of newly independent raccoons occurring in the fall months (September to November) [4]. In our models, raccoons born each year became independent by Week 38 (i.e., the second week of September or 20 weeks after the birth pulse in Week 18). After reaching independence, juvenile raccoons could contract and transmit rabies and disperse between Weeks 38 and 43. Dispersal leads to increased interactions and further geographic spread of the disease by infected individuals, thus causing a peak in new infections. Winter has previously been suggested to be a peak season for raccoon rabies infections, with increased social contact duration due to co-denning and thermoregulation [21]. However, peaks in raccoon rabies cases have been recorded in the spring and summer months in enzootic areas, with spring peaks possibly due to increased denning and contacts in the winter months and summer peaks possibly due to increased contact from juvenile dispersal [22]. In Southern Ontario, summer to autumn (July to September) was when rabies control efforts were undertaken, or are predicted to be completed in future years, between 2015 and 2025. In the United States, August to October was when similar programs were carried out to target newly susceptible juveniles [21]. We suggest that rabies control programs in Ontario should continue targeting juveniles in the late summer to early autumn to intercept with dispersing juveniles most effectively before reducing foraging activity in the colder months.

Another factor that affected rabies infections in raccoons in our models was the level of protective immunity in the raccoon population achieved by each intervention type. While seroprevalence of rabies antibodies obtained from field studies conducted by MNRF provided estimates of protective immunity associated with each intervention type, seroprevalence rates in the urban (6–14%) and rural (16–37%) habitats characterizing the intervention zone were very low. Using these low observed seroprevalence rates as a direct measure of the level of protective immunity in the model did not result in effective control of rabies, resulting in epicurves dissimilar to that of the observed raccoon rabies outbreak in Southern Ontario, while a seroprevalence 2.1 times higher than the measured rate resulted in epicurves closest to the actual observed infection trends (Appendix A). While surveillance data on rabid raccoons can be spatially and temporally biased, we used these reported case numbers as a guide to the general trend that was likely occurring at a broader scale. In our model, urban seroprevalence was therefore set to 23.1% (measured seroprevalence was 11%), while rural seroprevalence was set to between 42 and 50.4% (measured seroprevalence was 20% and 24%). A level of protective immunity of at least 60% has been previously suggested to be required to prevent an outbreak of raccoon rabies when tested on synthetic landscapes with varying quality and spatial heterogeneity of habitats [7]. These previous findings, coupled with our calibration results, could indicate that current serology testing is not sensitive enough or may have titres set at levels too conservative to reflect true immunity achieved by vaccination strategies.

Differences between the real epicurve and that predicted by the model indicate other factors may have also played a role in raccoon rabies case numbers, such as undetected cases (e.g., unreported culling of rabid raccoons by individuals). Uncertainties also exist regarding the effects of spatial variation in raccoons and the efficacy of disease control, as well as other factors difficult to measure in the field, such as heterogeneity in rates of rabies transmission between raccoons [7]. For the purposes of comparing the new SamPy model to the ORM, most biological and epidemiological parameters used were taken from previous ORM inputs for raccoon rabies. Other factors, such as start week, were determined based on calibration. However, it is also possible the outbreak occurred earlier than we predicted in our model. Nadin-Davis et al. [3] used phylogenetic analyses to suggest the outbreak may have occurred undetected as early as 2013. If so, raccoon rabies cases would have been spreading for two years prior to the starting week of infections in our model and may have been introduced in a location other than Hamilton. An earlier outbreak start date could explain how the disease appeared so well-established only a few weeks after its first recorded detection. However, our model was calibrated on and limited by the raccoon rabies data available. Despite differing suggested start times of the outbreak between our model and previous research, our model accounts for the changes in infections over time as of the time of the first detected case (Appendix A).

Due to the stochastic nature of agent-based modelling, we repeated simulations for each scenario 1000 times. Each simulation presented a potential reality, with differences in the physical and social profile of each raccoon agent that would ultimately affect disease outcomes. To account for these differences, random number generation was required, where each model had to be run starting with a random seed value. We used the same random seed value for each simulation across all scenarios, allowing runs to be reproducible and only allowing variation to occur when vaccines were applied (i.e., by starting with the same random seed value, the same sequence of random numbers would be generated for probability functions, such as the direction each raccoon would move when entering an adjacent cell, or the probability of natural death). This explains why 18.2% of simulations for each scenario resulted in rabies dying off within the first week. During this first week, no control interventions were applied. Agent mortality was therefore based solely on the probability distribution provided to the model, and by starting with the same seed value, this mortality was repeated across all simulations. This also meant that while Scenarios 1 and 2 showed rabies successfully being eliminated in 67.4% and 60.3% of simulations by 2025 and 2020, respectively, only 49.2% and 42.1% of these simulations predicted rabies elimination due to control interventions by 2025 and 2020, respectively.

We expect that the frequencies of successful raccoon rabies elimination following control interventions are underestimates, since our simulations did not allow for adaptive control strategies such as the shifting or intensification of control measures that would normally be carried out in response to changes in rabies distributions and case numbers. Furthermore, we ended simulations in 2025 because this was originally the predicted year for raccoon rabies elimination in Southern Ontario (MNRF, 2022). However, if raccoon rabies persists by the end of 2025 then the control programs will likely be continued until elimination is achieved. It is also likely that there are cumulative effects of two control methods being applied to the same area at the same time, but these effects are still unknown. In the case where a cell in our synthetic landscape received two controls in the same week and year, we applied the control with the higher seroprevalence, not a cumulative seroprevalence.

At the time of writing, raccoon rabies infections have still been reported by the MNRF as recently as December 2022 in the Niagara region, less than 50 km east of the first detected case (MNRF 2022). Even in simulations where control interventions were used until 2025, there was a 32.6% chance the disease would not be eliminated by 2025 and a low (<3%) chance of the disease spreading into the Niagara region. It is possible that we currently fall within this unlikely reality. It is also possible that our models have not accounted for other factors that may explain the persistence of raccoon rabies. For instance, our model did not account for other raccoon rabies hosts such as striped skunks (*Mephitis mephitis*) that are known to transmit raccoon rabies to other skunks and to raccoons [23]. Since 2015, 176 out of 517 (34%) of all raccoon variant cases in Ontario have been in skunks (MNRF 2022). An urban-centric outbreak of raccoon rabies is further complicated by more potential spillover hosts (e.g., larger concentrated populations of dogs and cats), which we could not account for in our model. Future modelling work should include a multi-species approach in order to evaluate the effect of striped skunks and other potential hosts on raccoon rabies persistence.

The urban-centric nature of this outbreak poses many challenges for rabies control programs in Southern Ontario. Common control methods for rural areas, such as the use of low-flying fixed-wing aircraft to distribute oral baits, cannot be used in urban settings. More people and domesticated animals also come into contact with baits, requiring additional coordination amongst control program officials to advise the public about coming into contact with baits. While these programs are expensive to implement, failure to do so can be even more costly in public health and economic ramifications. Our model suggests that the failure to use control interventions would lead to more than an 80% chance of the outbreak persisting until at least the end of 2025. Using control interventions until at least 2020 would cut this probability in half, and this prediction is likely underestimating the efficiency of the control programs due to adaptive disease management. In addition, our models confirm the strategic importance of control placement just north of Hamilton to prevent the passage of raccoons into the densely populated Greater Toronto Area where habitats are even more suitable for raccoons and where more humans and domestic animals are put at risk. Our study provides a valuable illustration of how agent-based modelling can be applied to evaluate potential outcomes of rabies management and help identify the most cost-effective approaches for rabies control, with widespread potential applications.

## Figures and Tables

**Figure 1 viruses-15-00528-f001:**
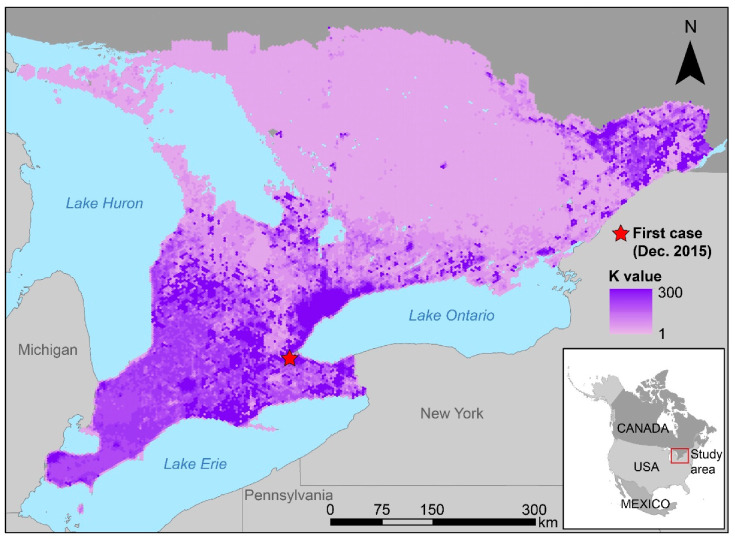
Map of the study area of Southern Ontario with K values shown with the location of the first detected raccoon rabies case. K values represented the predicted number of raccoons sustained by the habitat in each 10-km^2^ grid cell. A red star denotes the location of the first detected case of raccoon rabies in December 2015. The red square in the inset map denotes the study area.

**Figure 2 viruses-15-00528-f002:**
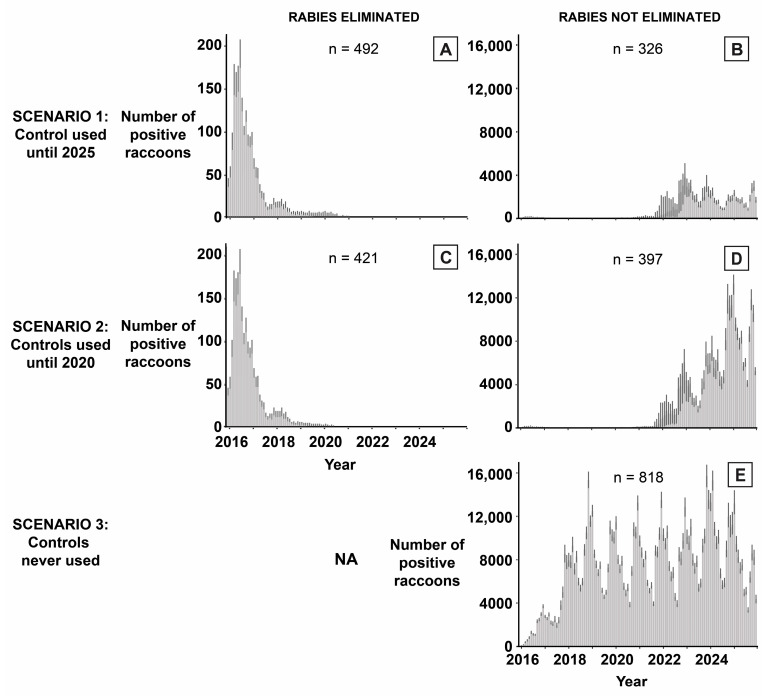
Epicurves showing the predicted median number of new infections of raccoon rabies per month for Scenarios 1, 2, and 3. Median case numbers (grey bars) are shown from November 2015 to December 2025 with upper and lower quartiles (dark lines). Panels (**A**,**C**) summarize simulations predicting rabies elimination in Scenarios 1 and 2, respectively, and (**B**,**D**,**E**) summarize simulations predicting rabies was not eliminated in Scenarios 1, 2, and 3, respectively. The number of simulations summarized per epicurve is shown above each panel. For Scenario 3, rabies was only predicted to be eliminated when the disease died off naturally within the first week following its introduction. Therefore, the epicurve for those simulations was not shown. Due to the scale of the y-axis, cases between November 2015 and December 2020 are not visible.

**Figure 3 viruses-15-00528-f003:**
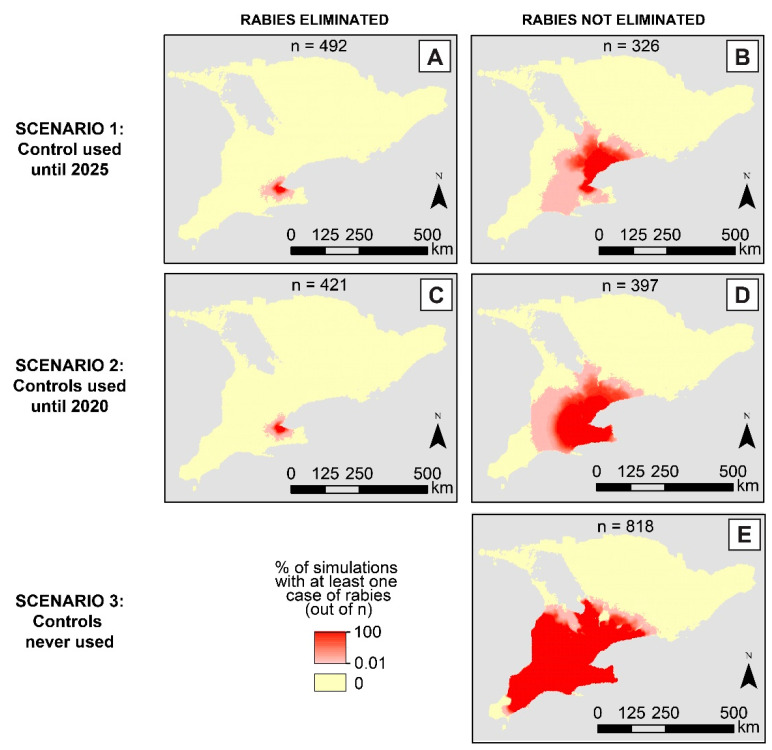
Maps showing the percentage (%) of simulations that predicted at least one raccoon rabies case per 10-km^2^ grid cell between November 2015 and December 2025. Panels (**A**,**C**) summarize simulations predicting rabies elimination in Scenarios 1 and 2, respectively, where simulations with rabies die off in the first week were removed (*n* = 182). Panels (**B**,**D**,**E**) summarize simulations predicting rabies was not eliminated in Scenarios 1, 2, and 3, respectively. The number of simulations used to calculate percentages per map is shown above each panel. For Scenario 3, rabies was only predicted to be eliminated when the disease died off naturally within the first week following its introduction. Therefore, the map for those simulations was not shown.

**Table 1 viruses-15-00528-t001:** SamPy parameters and inputs used for the raccoon population growth and rabies dynamics.

Raccoon Population Growth Parameters †	
**Definition of Parameter**	**Input Settings**
Number of initial breeding pairs	5 (5 males, 5 females)
Age of breeding raccoon agents	52 weeks
Number of simulation years for population building	150 years
Age of independence (i.e., age when raccoon agent becomes a juvenile)	20 weeks
Age adulthood is reached	75 weeks
Gestation period	9 weeks
Week of the year in which females give birth	Week 18
Annual mortality rates	Probability by age (0; 1; 2; 3; 4; 5; 6; 7) For both sexes (0.6; 0.4; 0.3; 0.3; 0.3; 0.6; 0.6; 0.6)
Mean litter size	4
Litter size variance	1
Probability density values for number of offspring produced in a litter	Number of offspring possible (0, 1, 2, 3, 4, 5, 6, 7, 8, 9) Probabilities (0.0001, 0.0044, 0.0539, 0.2416, 0.4000, 0.2416, 0.0539, 0.0044, 0.0001)
Litter sex ratio	50
Period of dispersion (week(s) of the year, where juveniles are defined as 20 weeks or older but under 75 weeks of age)	Juvenile males: 38–43 Juvenile females: 38–43 Adult males: 8–43 Adult females: 8–15 and 38–43
Distance of dispersion	The probability of displacement by a particular distance (number of grid cells: probability) Juvenile males (0: 75.1, 1: 12.9, 2: 6.5, 3: 1.8, 4: 0.9, 5: 0.9, 6: 0.92, 7: 0, 8: 1, >=9: 0) Juvenile females (0: 90.8; 1: 4.07; 2: 1.4, 3: 0.3, 4: 0.7, 5: 0.3, 6: 1.02, 7: 0, 8: 1, >=9: 0) Adult males (0: 88.9; 1: 4.25; 2: 1.4; 3: 0.9; 4: 0.7, 5: 0.9, 6: 0, 7: 1, 8: 1, >=9: 0) Adult females (0: 92.3, 1: 3.06, 2: 1, 3: 0.9, 4: 0.4, 5: 0.7, 6: 0.73, >=7: 0)
Probability of leaving current grid cell to interact with raccoons in any of the six neighbouring grid cells	22.2%, divided equally between neighbouring grid cells
Rabies dynamics parameters	
Chance of death from raccoon rabies	95%
Transmission probability (the probability that pathogen transmission will occur given contact)	3.5%
Level of initial infection	5%
Week when rabies was introduced	Week 47 of 2015
Incubation period	Number of weeks of infection: Probability of incubation period lasting x number of weeks (1: 1, 2: 5, 3: 5, 4: 10, 5: 15, 6: 20, 7: 15, 8: 10, 9: 5, 10: 5, 11: 5, 12: 2, 13: 1, 14: 1)
Infection period	100% chance of infectious period lasting 1 week

† Inputs for biological parameters are based on previous work by [7].

## Data Availability

The rabies case data presented in this study are available on the Canadian Food Inspection Agency website (https://inspection.canada.ca/animal-health/terrestrial-animals/diseases/reportable/rabies/rabies-in-canada/eng/1356156989919/1356157139999, accessed 22 December 2022) and Ministry of Natural Resources and Forestry website (https://www.ontario.ca/page/rabies-cases, accessed 21 December 2022).

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
