# Peer review of "Comparing Control Intervention Scenarios for Raccoon Rabies in Southern Ontario between 2015 and 2025"

_viruses, 2023, doi:10.3390/v15020528_

Round 1
Reviewer 1 Report
Comments and Suggestions for Authors
Authors Acheson et al. presents an agent based model to evaluate the control measures applied to an urban outbreak of raccoon rabies. The model was comprehensively and clearly described along with reasoning for the approach employed. The paper is very well written and provides excellent coverage of limitations and discussion around future use of the model. The below are suggestions to add some clarity at certain points:
Line 65: the word available seems unclear here. What exactly available? Just the bait stations?
Line 164 to 166: By setting the mortality to 95% due to naturally occurring rabies antibodies, is the assumption that naturally occurring rabies antibodies are protective? If so, is there evidence that this is the case?
Line 198 to 202: Why is there a difference in period of immunity between raccoons receiving a vaccine and raccoons surviving rabies?
Author Response
Please see the attachment for responses to Reviewer 1.

Reviewer 2 Report
I appreciate the opportunity to review this manuscript on the use of a model to evaluate the efficacy of rabies control programs instituted in response to an outbreak of raccoon rabies in an urban/suburban environment. This is a thoughtful and well-written manuscript and I have a limited number of comments for consideration by the authors.
Line 39-40: It may be accurate that this model could be used to plan interventions but the paper only varies whether interventions where used and how long they were applied for. Planning an intervention implies more strategy about where and when and which method. I think in this instance it is more accurate to say evaluate an approach rather than plan for intervention.
Line 53: recommend not using the plural variants since the variant on the eastern seaboard is the raccoon variant (and bats) but not other terrestrial variants
Line 142: Please clarify whether direction was stochastic or random
Line 151: Why was 5% chosen?
Line 157-158: In reality, my experience with rabid raccoons suggests that their activity patterns would be reduced compared to healthy animals.
Lines 183-186 and Appendix S2: The greater incidence of rabies in Figure S2.1 B later in the outbreak compared to the real outbreak suggests that a difference in seroprevalence does not completely explain the variability between the modeled curve and the epi curve.
Table 1: There are input settings which I did not find referenced anywhere in the paper and it is not clear how they were chosen. Example: annual mortality rates by age, distance of dispersion
Lines 454-455: The explanation provided for the finding that the model is accurately identifying a peak in infections in the winter when observed infections peak in the summer seems plausible but in warmer areas with established raccoon rabies, declines in both numbers and percent positivity are observed in the winter.
Lines 474-475: As previously mentioned, the idea that the true seroprevalence is not being accurately measured is certainly one possibility but it isn't clear what other changes to inputs authors tried that might also have produced a modeled epi curve that more closely resembled the observed one.
Author Response
Please see the attachment for responses to Reviewer 2.
